# Identification of Risk Factors and Production Practices Associated with Type 2 Porcine Reproductive and Respiratory Syndrome Virus (PRRSV) Positivity on Pig Farms in Jalisco, Mexico

**DOI:** 10.3390/pathogens14090881

**Published:** 2025-09-03

**Authors:** Alberto Jorge Galindo-Barboza, José Francisco Rivera-Benítez, Jazmín De la Luz-Armendáriz, José Iván Sánchez-Betancourt, Jesús Hernández, Suzel Guadalupe Sauceda-Cerecer, Jaime Enrique De Alba-Campos

**Affiliations:** 1Programa de Doctorado en Ciencias de la Producción y de la Salud Animal, Universidad Nacional Autónoma de Mexico, UNAM, Mexico City 04510, Mexico; aljogaba@gmail.com; 2Laboratorio de Virología, Centro Nacional de Investigación Disciplinaria en Salud Animal e Inocuidad, Instituto Nacional de Investigaciones Forestales, Agrícolas y Pecuarias, INIFAP, Cuajimalpa, Mexico City 05110, Mexico; 3Facultad de Medicina Veterinaria y Zootecnia, Universidad Nacional Autónoma de Mexico, UNAM, Mexico City 04510, Mexico; delaluzarmendarizj@fmvz.unam.mx (J.D.l.L.-A.); ivan.sanchez@posgrado.unam.mx (J.I.S.-B.); 4Laboratorio de Inmunología, Centro de Investigación en Alimentación y Desarrollo, A.C., Hermosillo 83304, Sonora, Mexico; jhdez@ciad.mx; 5Grupo Estatal de Vigilancia Epidemiológica, Comité Estatal para el Fomento y Protección Pecuaria del Estado de Jalisco, El Salto 45690, Jalisco, Mexico; suzelsauceda@gmail.com; 6Unión Regional de Porcicultores de Jalisco, El Salto 45680, Jalisco, Mexico; presidencia@urpj.org.mx

**Keywords:** porcine reproductive and respiratory syndrome virus (PRRSV), production practices, Jalisco, Mexico

## Abstract

The modernization of pig farming has resulted in increasingly dense pig populations. While this accelerates production and ensures a steady pork supply, it also increases the risk of infection transmission. As an endemic and widely distributed pathogen, porcine reproductive and respiratory syndrome virus (PRRSV) type 2 can cause disease, depending on the production practices implemented. This study evaluated pig production conditions in Jalisco, Mexico, as well as how these conditions correlated with PRRSV detection. In total, 4207 serum samples obtained from 80 pig farms were subjected to analysis, and epidemiological information was collected to evaluate potential determinants of PRRSV presence. Positive samples were most frequently found in pigs up to 10 weeks of age, raised in semi-intensive, intensive, farrow-to-finish farm, and multisite systems, with relative frequency values ranging from 58.6% to 76.9%. The results revealed that various production practices, particularly related to biosecurity protocols, were associated with the presence of PRRSV on the farms evaluated.

## 1. Introduction

Porcine reproductive and respiratory syndrome (PRRS) is a viral disease that affects the majority of pig populations worldwide, although some countries, such as Australia, New Zealand, India, Chile, and the Nordic countries, remain virus-free [1]. The clinical presentation of PRRS varies considerably, due to multiple factors including the environmental conditions [2,3], immune status [4], genetic characteristics of the virus and host, both of which contribute to susceptibility and pathogenesis [5], production unit size, management practices (including vaccination) [6], biosecurity measures [7,8], and coinfection with other viruses and secondary pathogens present on the farms [9].

In the USA, total industry losses due to PRRS in 2011 were estimated at USD 664 million, with 45% of the losses reported on breeding farms [10]. In Mexico, PPRSV has increased production costs by 4.7% to 35.9% on farrow-to-finish commercial farms compared with PPRSV-negative farms. It is estimated that PRRS causes annual losses of USD 80–120 million [11].

PRRS is caused by a virus (PRRSV) belonging to the family *Arteriviridae* within the order *Nidovirales*. It is an enveloped positive-sense single-stranded RNA virus with a genome of approximately 15 kb. PRRSV is classified into two genotypes, type 1 (PRRSV-1) and type 2 (PRRSV-2), with approximately 44% nucleotide sequence variation between them. Its genome is organized into 10 open reading frames (ORFs): 1a, 1b, 2a, 2b, 3, 4, 5, 5a, 6, and 7 [12]. ORFs 1a and 1b constitute approximately 80% of the genome and encode 12 nonstructural proteins involved in recombination, transcription, and replication. ORFs 2a, 2b, 3–7, and 5a encode the structural glycoproteins GP2a, GP3, GP4, and GP5, as well as the non-glycosylated proteins 2b, the M membrane protein, and the N nucleocapsid protein [13].

In 2022, Mexico ranked eighth worldwide in pig production, holding a total inventory of 20,894,799 animals, according to the most recent official data [14]. The pork industry represents a key contributor to the national economy, with Jalisco leading the country in production by generating 397,849.4 tons of pork from 4,797,931 pigs, accounting for 23% of the national inventory in 2022 [14]. This region is characterized by modern farming infrastructure, intensive swine production systems, and a well-established capacity for pork export. However, the industry also includes approximately 1800 small-scale operations, such as family-run and backyard farms, which account for about 58% of the farms in Jalisco. This diversity highlights the challenges of standardizing and implementing disease control measures across the sector [15].

Limited information exists in Mexico regarding the relationship between PRRSV infection rates and farming practices. This study used Jalisco state as a case study to explore the associations between production practices and the prevalence of PRRSV. Furthermore, this study aimed to develop management and monitoring strategies for PRRSV based on a methodological framework that can inform similar efforts in regions with diverse production scales, population densities, and biosecurity practices, as exemplified by the swine industry in Jalisco.

## 2. Materials and Methods

### 2.1. Research Site

The research was conducted on pig farms situated in Jalisco, Mexico. According to the Unión Regional de Porcicultores de Jalisco (URPJ), the state is divided into four zones—A, B1, B2, and B3—based on pig population density [15]. The density of pigs in these regions is 9.46, 210.36, 261.65, and 135.4 pigs per km^2^, respectively [16].

### 2.2. Samples

We used serum samples from a previous study [16], comprising 4207 specimens collected from 80 farms. Farms and samples were organized into four clusters corresponding to the previously defined regions. Analyses were conducted at the Virology Laboratory of INIFAP (Palo Alto, Mexico City, Mexico).

Farms were classified according to the number of reproductive sows into semi-intensive farms (21–500 sows) and intensive farms (≥500 sows). They were also categorized by production system: farrow-to-finish farms (FF) and multisite farms (MS). Additionally, animals were grouped by age: 0–3, 4–10, 11–14, 15–18, and 19–22 weeks, as well as pregnant sows.

A maximum of 60 serum specimens were obtained from each farm, representing the production stages present on the farm, with up to 10 specimens collected per stage. Samples were taken from pigs both exhibiting and not exhibiting clinical signs suggestive of PRRS infection. Of the total 4207 specimens, 3802 came from asymptomatic pigs, while 405 were from pigs showing clinical signs. Serum samples were pooled by production stage, with five specimens per pool and two pools per stage. In farrow-to-finish (FF) farms, a total of 12 pools were obtained. In multisite (MS) farms, pools were created only for the stages relevant to their specific production purpose. Overall, 844 pools were generated and identified according to the previously established clusters.

All serum samples from the farms were included in the study, regardless of the PRRS vaccination status. The farm managers reported that only modified live virus (MLV) vaccines were used, and the vaccination protocol focused on breeding sows, with administration at regular intervals of approximately 4 months, and on replacement gilts, 2–4 weeks prior to their first insemination. Additionally, the vaccination of piglets typically occurred between 4 and 10 weeks of age; however, this practice was eventually implemented in only 2 of the farms included in the study.

The distribution of pigs, samples, and pools by age and region, as reported in our previous study [16], formed the basis of this analysis. Detailed information on the number of pigs sampled per age and region is available in the referenced publication.

### 2.3. RT-qPCR

Detection of ORF7 was carried out using the QuantiTect^®^ Probe RT‒PCR Kit (QIAGEN, Hilden, Germany, Cat. No. 204445) in a 10 µL reaction, containing 5 µL of 2× Master Mix, 2 µL of RNA (≤500 ng/reaction), 1.03 µL of RNase-free water, and primers and probe at 0.4 µM each, following Kleiboeker et al. [17]. Reverse transcriptase quantitative real-time polymerase chain reaction (RT-qPCR) was performed on a CFX96™ Real-Time System (Bio-Rad Laboratories, Hercules, CA, USA), with cDNA synthesis at 50 °C for 30 min, an initial denaturation at 95 °C for 15 min, and 40 amplification cycles of 94 °C for 15 s and 60 °C for 1 min. Fluorescence data were analyzed using Bio-Rad CFX Manager 3.1 software with drift corrections to determine the cycle threshold (Ct). Positive controls ensured consistency, and samples with Ct values above 35 were classified as negative.

### 2.4. Survey Data and Risk Factor Assessment

A survey developed by our team was applied to identify potential risk factors associated with PRRSV detection in the collected samples. The questionnaire assessed multiple aspects of farm management, including biosecurity measures, feed and water provision, facility standards, and waste management. Detailed categorization of these factors and their definitions were previously published and can be consulted [16].

### 2.5. Statistical Analysis

Data from the survey of 80 farms were organized into a database for statistical analysis. Cluster analysis was conducted using 2 × 2 contingency tables, applying the chi-square (χ^2^) test at a significance level of α = 0.05 to assess independence between variables. Yates’ correction was applied for cells with fewer than ten observations. For variables showing a significant association (*p* < 0.05), odds ratios (ORs) and 95% confidence intervals (CIs) were calculated. Variables not associated with PRRSV presence in any stratum were excluded from further analysis.

Reference categories for OR calculations were defined based on the risk factor groups. Establishing these reference groups allowed for quantitative comparisons and facilitated interpretation of both the magnitude and direction of associations. This approach ensures that the OR accurately reflects the relative likelihood of the outcome, providing a robust basis for interpreting the results.

All statistical analyses were conducted using RStudio (version 2023.06.0) and R (version 4.3.1), employing the “epibasix” package for key epidemiological and biostatistical functions.

## 3. Results

Among the farms included in this study, 95% (76/80) were positive for PRRSV, with at least one positive sample detected per farm. Analyses were then conducted on pooled samples from different production stages across all the farms. The relative frequency (RF) of PRRSV-positive pools measured by RT-qPCR was 44.6% in B2, 35.1% in B3, 34.5% in A, and 30.7% in B1.

Table 1 presents the RFs of PRRSV by age group of pigs and by farm type, categorized according to the number of breeding sows. For the semi-intensive pig farms studied, an RF of 36.2% was observed, with the highest RF recorded in Region B2 at 52.2%. Conversely, intensive pig farms presented an RF of 36%, with Region B1 showing the highest frequency at 50%.

Farrow-to-finish (FF) farms accounted for 85% of the farms included in the study, while multisite (MS) farms represented the remaining 15% (5 in site 1 and 7 in site 3). The overall RF of PRRSV was 36.2% in FF farms and 35.6% in MS farms. In FF farms, A Region had an RF of 34.1%, B1 Region 29.4%, B2 Region 46%, and B3 Region 36.9%. Among MS farms, site-1 farms exhibited an RF of 44.4%, while site-3 farms showed an RF of 30.4%.

A total of 66.2% of the farms (53/80) reported using PRRSV vaccination, although its adoption varied across regions. In Region A, which has the lowest pig density, the per-centage of vaccination coverage was 55.9%, whereas in other regions, it reached 70% (Region B1) and up to 72.2% (Regions B2 and B3).

Table 2 summarizes the potential risk factors for PRRSV, as determined by chi-square analysis. The odds ratios (ORs) indicate the relative probability of PRRSV detection in farms exposed to each risk factor compared with those unexposed.

## 4. Discussion

The relationships between farm-level biosecurity practices, potential risk factors, and PRRSV presence were explored using viral RNA detection via RT-qPCR (hereafter referred to as viremia). Significant links between specific risk factors and viremia were identified through chi-square (χ^2^) tests and odds ratios (ORs). The key findings are summarized and discussed in the following sections.

Establishing efficient control measures against PRRSV largely depends on prior knowledge of the health status and management practices implemented in the production units. This knowledge enables the identification of opportunities to improve the sanitary status of farms and, ultimately, enhance the productivity. The regionalization currently used in Jalisco, although initially designed for animal health surveillance, provides a justified segmentation based on pig density per/km^2^ [15,16]. This approach has allowed for the correlation of animal density with the incidence of diseases such as PRRSV. A high prevalence of PRRSV infection was observed in Jalisco, with 95% of the evaluated farms testing positive. Of the 53 farms that had been vaccinated, only two were negative by RT-qPCR, while the remainder showed evidence of infection. Similarly, among the 27 non-vaccinated farms, two tested negative, and the rest tested positive. In contrast to the findings reported by Espinosa et al. [15], where only 12.4% of commercial farms and back-yard farms in Jalisco used vaccination as a control measure, the adoption of this strategy in the evaluated farms appears to be high. However, the relative frequency of viremia remains significant, despite sampling pigs that showed no clinical signs and appeared healthy. This highlights the need for further studies to determine whether viremia is associated with vaccination or results from infection with field strains.

Moreover, all semi-intensive farms across the four regions and the intensive farms in Regions A and B1 tested positive. In regions B2 and B3, 91.7% and 57.1% of intensive farms, respectively, were also positive for PRRSV. A similar pattern was observed for far-row-to-finish and multisite farms, indicating widespread viremia across production systems.

Region B2 accounts for the largest proportion of pigs in Jalisco’s swine inventory, representing the area with the highest swine density in the state. In line with this, it presented the highest frequency of positive pooled samples (44.6%) among all the regions analyzed. A similar pattern has been observed for other swine diseases, such as those caused by PCV2 [16], suggesting a potential association between high animal density and the prevalence of viremia.

The production stages with the highest RF of viremia were the early stages, from birth to weaning (38.2%), from weaning to 10 weeks of age (61.1%), and from 11 to 14 weeks (40.4%), indicating a significant postweaning challenge. This trend was also observed for semi-intensive (58.6%), intensive (68.4%), farrow-to-finish (59.6%), and multisite farms (76.9%). Although the pig producers did not provide specific information on the commercial brands of vaccines used or the vaccination protocols implemented, it is estimated—based on a previous study [18] and unpublished census data from the local swine producers’ association in Jalisco—that approximately 90% of farms that vaccinate do so in the breeding stock. Therefore, it can be inferred that the viremia detected in the previously mentioned stages likely corresponds to field-strain infections. Moreover, according to the literature, pigs in these production stages are more susceptible to PRRSV infection due to the immunological stress caused by environmental challenges and the continuous physiological and management transitions (e.g., changes in housing, diet, or regrouping) [19]. Additionally, the high percentages of PRRSV-positive samples in suckling piglets (ranging from 36.1% to 44.4%) suggest ongoing viral circulation among breeding sows, likely due to exposure to either vaccine strains or wild-type field strains. Therefore, it is essential to design field studies aimed at identifying infection sources and developing effective mitigation strategies adaptable to most farms.

In regions where PRRSV is endemic, estimating the prevalence of field-virus infection is challenging, and available data are limited. However, approximately 60–80% of herds are typically infected [19]. The results of this study fall within the expected range and present the observed frequencies of PRRSV positivity across the analyzed strata.

Given the above findings, identifying the main risk factors associated with viremia is crucial. In the disinfection practices and fomite control category, the lack of access control to farms in region B3 and semi-intensive farms were associated with 7.6- and 6.4-fold greater odds of being positive for viremia, respectively, than those implementing access control measures. In other regions and farm types, this factor was not associated with viremia due to its common implementation as a standard practice. In region B3, biosecurity measures are often relaxed because of the medium-to-low population density and the predominance of semi-intensive production. However, this region serves as a transition zone with frequent pig movements, which poses significant risks.

Another risk associated with viremia is when the technical staff visits other farms, a predominant situation in Region A, particularly farrow-to-finish farms, where higher odds of viremia (1.6–1.7) are observed than farms with an exclusive veterinarian. The presence of viremia indicates the potential recirculation of the virus within the farms. Protocols exist, where even in farms located in high-density swine areas, in addition to PRRSV vaccination, it is crucial to limit the movement of personnel in and out of farms to produce PRRSV-negative pigs at weaning [20], the stage with the highest frequency of positive samples.

In the sanitary management category, the absence of deworming protocols was associated with 2.4- and 2.5-fold greater odds of viremia in pigs from semi-intensive and farrow-to-finish farms, respectively, than in those from farms with established deworming protocols. Tissue damage caused by certain parasitic infections can exacerbate pneumonia caused by bacteria and viruses such as PRRSV. Additionally, parasitic infections can impair nutrient absorption, complicating clinical outcomes [21]. Furthermore, some parasites, such as Ascaris suum, can compromise immune responses, perpetuating viral infections [22,23].

Other sanitary management risks observed in the farms studied include the absence of immunizations against various endemic diseases. The lack of protection against Bordetella/Pasteurella, *Glaesserella parasuis*, Circovirus, and Erysipela increases the likelihood of PRRSV viremia compared with pigs with vaccination schemes against these pathogens.

In pigs aged between 15 and 18 weeks who were not vaccinated against *G. parasuis* or Erysipela, the odds of developing PRRSV viremia were 3.4 and 3.8 times greater, respectively, than in vaccinated pigs. Similarly, pregnant sows who were not vaccinated against *G. parasuis* were 4.8 times more likely to be PRRSV viremic. However, the Pearson chi-squared test showed no statistically significant association between PRRSV vaccination in pregnant sows and the presence of viremia. Previous studies have reported that G. parasuis infection increases the number of PRRSV copies in nasal secretions, blood, and lung tissues, which, along with the associated inflammatory response, exacerbates clinical signs [24]. In pregnant sows, the onset of the porcine respiratory disease complex (PRRSV—*G. parasuis*) could lead to increased sow mortality [25].

In Region B2, the area with the highest swine density, the risk factor “circo”, was as-sociated with 3.6 odds of PRRSV viremia. This is particularly relevant, as a previous study conducted by our research team estimated a 75% prevalence of PCV2-positive farms in Jalisco with a vaccination coverage of 88.7% in the farms studied [15], and in the same region, an FR of 66.7% was observed [16]. These findings further suggest a correlation between high swine density and infection and viremia caused by both PRRSV and PCV2.

The absence of PRRSV vaccination was not identified as a risk factor in Region B3 or in piglets during the weaning-to-10-week stage (strata in which a statistically significant association was observed using the chi-square test). The odds ratios (OR) observed in these groups were 0.28 (95% CI: 0.10–0.78) and 0.26 (95% CI: 0.13–0.55), indicating that the lack of vaccination was associated with a lower likelihood of viremia. By inverting the reference categories in the OR calculation, reciprocal values were obtained, allowing an alternative interpretation of the effect of vaccination on the probability of viremia. This analysis showed that pigs from farms implementing the vaccination program had higher odds of developing viremia compared to pigs from non-vaccinated farms, with OR of 3.6 (95% CI: 1.3–10) in Region B3 and 3.8 (95% CI: 1.8–7.7) in piglets from weaning to 10 weeks.

This finding may be related to the characteristics of the vaccines used on the studied farms. According to the data collected, the vaccination coverage in Region B3 reached 72.2%, and the vaccination of piglets typically occurred between 4 and 10 weeks of age, although the practice of vaccinating piglets is very low, only 2.5% (2/80) of farms reported having done so during the study period, and not consistently. The use of modified live vaccines (MLVs), which are known to induce transient viremia postvaccination [26], could partially explain the observed association. In contrast, unvaccinated pigs, although susceptible to infection, may not have been exposed to the virus during the study period and thus did not exhibit detectable viremia. Importantly, the statistical association between vaccination status and viremia was significant for both the B3 Region (χ^2^(1, N = 165) = 5.49, *p* = 0.0191) and the weaning to 10-week stage (χ^2^(1, N = 146) = 12.14, *p* = 0.0004), highlighting the need for further investigation into the dynamics of vaccine-derived and field-strain viremia in vaccinated populations.

Importantly, the risk factors not included in Table 2 did not show significant differences based on the data collected. However, this does not imply that these factors are irrelevant for PRRSV dissemination. For instance, the implementation of the “all-in/all-out” system is a key measure to prevent PRRSV transmission and was reported in 94.3% of farms in a U.S. study, contributing to risk reduction [6]. In the present study, although the observed viremia could primarily result from vaccination, some cases may reflect field-virus infection, particularly when farm access control is compromised, affecting biosecurity. Previous research has shown that strict adherence to biosecurity recommendations is essential to achieve infection stabilization within the farm [20].

Therefore, it is crucial to consider each of these and assess their impact on each production unit, with the aim of preventing viral recirculation and, consequently, reinfections that compromise the productive capacity of farms.

## 5. Conclusions

This study highlighted the key production practices associated with PRRSV viremia in pig farms in Jalisco, Mexico. Evaluating prevalence across different production systems allowed the identification of critical intervention points related to sanitation, disinfection, and fomite management. Implementing targeted measures at these points can improve the control of PRRSV viremia and enhance overall swine productivity. However, the observed associations are derived from a cross-sectional study and do not imply causality. Future longitudinal studies are warranted to provide more precise and robust insights into the factors and practices influencing viral presence.

## Figures and Tables

**Table 1 pathogens-14-00881-t001:** Relative frequency of PRRSV-positive pools by age group and farm type (semi-intensive, intensive, farrow-to-finish, and multisite systems).

Age Range of the Pigs	Relative Frequency. Pools Positive/Analyzed.				
	All Pools	Semi-Intensive Pig Farming (21–500 sows)	Intensive Pig Farming (≥500 sows)	Farrow-to-Finish Farm (FF)	Multisite Farms (MS)
Birth to weaning	47/123 (38.2%)	34/87 (39.1%)	13/36 (36.1%)	43/114 (37.7%)	4/9 (44.4%)
Weaning to 10 weeks	91/149 (61.1%)	65/111 (58.6%)	26/38 (68.4%)	81/136 (59.6%)	10/13 (76.9%)
11 to 14 weeks	65/161 (40.4%)	47/119 (39.5%)	18/42 (42.9%)	59/149 (39.6%)	6/12 (50%)
15 to 18 weeks	42/152 (27.6%)	28/100 (28%)	14/52 (26.9%)	39/137 (28.5%)	3/15 (20%)
19 to 22 weeks	34/126 (27.0%)	25/82 (30.5%)	9/44 (20.5%)	33/112 (29.5%)	1/14 (7.1%)
Pregnant sows	26/133 (19.5%)	15/92 (16.3%)	11/41 (26.8%)	24/123 (19.5%)	2/10 (20.0%)
Total for All Pools	305/844 (36.1%)				

**Table 2 pathogens-14-00881-t002:** Association between potential risk factors and PRRSV detection across analyzed clusters. Chi-square (χ^2^) statistics and odds ratios (ORs) are shown, indicating the relative likelihood of PRRSV presence in farms exposed to each risk factor compared with unexposed farms in Jalisco, Mexico.

Risk Factors	Group	χ^2^	Obs.	*p*	Odds Ratio	Confidence Interval 95%
access	B3-region	6.22	165	0.0126	7.6	1.53–38.14
	Semi-intensive	5.16	587	0.0230	6.4	1.31–30.94
technical	A-region	4.90	377	0.0268	1.6	1.08–2.53
	Semi-intensive	8.01	587	0.0046	1.7	1.19–2.43
	Farrow-to-finish farm	9.37	745	0.0022	1.6	1.2–2.24
deworm	Semi-intensive	4.71	587	0.0298	2.5	1.15–5.32
	Farrow-to-finish farm	4.66	756	0.0307	2.4	1.14–5.24
bp	B2-region	10.10	157	0.0014	3.2	1.59–6.33
parasuis	B2-region	7.91	157	0.0048	2.9	1.42–5.74
	Pigs aged 15 to 18 weeks	6.77	150	0.0092	3.4	1.4–8.44
	Pregnant sows	5.62	127	0.0177	4.8	1.35–17.2
	Multisite (Site 3)	4.59	433	0.0319	1.7	1.07–2.63
prrs	B3-region	5.49	165	0.0191	0.28	0.10–0.78
	Weaning-to-10-week	12.14	146	0.0004	0.26	0.13–0.55
circo	B2-region	4.81	157	0.0282	3.6	1.23–10.57
ery	Semi-intensive	8.23	587	0.0041	2.1	1.28–3.42
	Pigs aged 15 to 18 weeks	5.19	150	0.0226	3.8	1.26–11.62
fence	Farrow-to-finish farm	4.41	740	0.0357	1.9	1.08–3.38
wild	B2-region	6.94	149	0.0084	12.6	1.55–101.96

*p* = *p* value; Obs. = observations. Decoding risk factors by category: access: No farm access control; technical: technical staff visit other farms; deworm: Deworming not conducted; bp: lack of *Bordetella* and *Pasteurella* vaccination; parasuis: Lack of *Glaesserella parasuis* vaccination; prrs: lack of PRRSV vaccination; circo: lack of PCV2 vaccination; ery: lack of Erysipelas vaccination; fence: No perimeter fencing; wild: presence of wild pigs near farms.

## Data Availability

Farm data are restricted due to privacy regulations.

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
