# Peer review of "Identification of Risk Factors and Production Practices Associated with Type 2 Porcine Reproductive and Respiratory Syndrome Virus (PRRSV) Positivity on Pig Farms in Jalisco, Mexico"

_pathogens, 2025, doi:10.3390/pathogens14090881_

Round 1

Reviewer 1 Report

Comments and Suggestions for Authors

The manuscript provides a comprehensive and well-structured analysis of the risk factors and production practices associated with PRRSV in pig farms in Jalisco, Mexico. The study is methodologically sound, with clear objectives, robust data collection, and appropriate statistical analyses. The findings are highly relevant to the swine industry, particularly in regions with similar production systems and disease challenges.

I have some comments to the authors:

Introduction, doesnt run flow, needs to be re-written.

Family and Order of the virus should be italic.

Line 60-62: Rephrase please, its not clear.

Line 66: Add citation please.

Line 90: Samples, please rewrite, and make easier to understand, maybe a chart can be added to show how sampling can be done. 

The manuscript mentions that vaccination coverage was high (66.2% of farms), but the specific vaccines (e.g., MLV vs. killed) and protocols used are not detailed. This information is critical for interpreting the association between vaccination and viremia.

Suggest adding a table or supplementary material summarizing vaccination practices (e.g., timing, frequency, vaccine type) across farms.

Discussion of Limitations:

The cross-sectional design limits causal inferences. Acknowledging this and suggesting longitudinal studies for future research would strengthen the manuscript.

The reliance on pooled samples, while practical, may underestimate individual-level variation in viral load. This could be briefly discussed.

In the discussion section:

Compare the high PRRSV prevalence in Jalisco with other regions (e.g., the U.S. or Europe) to highlight regional differences or similarities.

Discuss how the identified risk factors align or contrast with findings from other studies (e.g., references [6], [7], [20]).

Check please for english language, and typos. 

Comments on the Quality of English Language

Needs english language editing, grammer and typos. Making the language run flow and easy.

Author Response

Thank you very much for taking the time to review this manuscript. Please find detailed responses below and the corresponding revisions/corrections highlighted/in track changes in the re-submitted files.

Comments 1: Introduction, doesnt run flow, needs to be re-written.

Response 1: Thank you for pointing this out. The introduction has been revised.

Comments 2: Family and Order of the virus should be italic.

Response 2: Agree. We have revised the manuscript accordingly to italicize the family (Arteriviridae) and order (Nidovirales) of the virus throughout the text.

Comments 3: Line 60-62: Rephrase please, its not clear.

Response 3: Thank you for pointing this out. We agree with this comment. Therefore, we have revised the paragraph to improve clarity and readability. The updated text now clearly describes the genome organization and the proteins encoded by each ORF.

Comments 4: Line 66: Add citation please.

Response 4: Thank you for pointing this out. We agree with this comment. Therefore, we have added a citation to support the statement regarding Mexico’s pig production in 2022.

Comments 5: Line 90: Samples, please rewrite, and make easier to understand, maybe a chart can be added to show how sampling can be done.

Response 5: Thank you for pointing this out. We agree with this comment. Therefore, we have carefully revised and reworded section 2.2 “Samples” to improve clarity and make the description of the sampling process easier to understand. All changes are marked in red in the revised manuscript.

Comments 6: The manuscript mentions that vaccination coverage was high (66.2% of farms), but the specific vaccines (e.g., MLV vs. killed) and protocols used are not detailed. This information is critical for interpreting the association between vaccination and viremia.

Response 6: Thank you for pointing this out. We agree with this comment. Therefore, we have added detailed information regarding the type of vaccine and the vaccination protocol used on the farms in the “Samples” section of Materials and Methods.

Comments 7: Suggest adding a table or supplementary material summarizing vaccination practices (e.g., timing, frequency, vaccine type) across farms.

Response 7: We thank the reviewer for this suggestion. However, we believe that additional tables or supplementary material are not necessary, as all farms followed a single, uniform vaccination protocol. The type of vaccine (MLV) and its timing—administered to breeding sows approximately every 4 months and to replacement gilts 2–4 weeks prior to first insemination—are fully described in the “Samples” section of Materials and Methods.

Comments 8: The cross-sectional design limits causal inferences. Acknowledging this and suggesting longitudinal studies for future research would strengthen the manuscript.

Response 8: We thank the reviewer for this suggestion. The Conclusion section has been revised to clarify that the observed associations are based on a cross-sectional design and do not imply causality. Furthermore, we highlight the recommendation for future longitudinal studies to provide more precise and robust insights into the factors and practices associated with PRRSV viremia.

Comments 9: The reliance on pooled samples, while practical, may underestimate individual-level variation in viral load. This could be briefly discussed.

Response 9: We appreciate your comment. This point has been addressed in the Discussion section.

Comments 10: Compare the high PRRSV prevalence in Jalisco with other regions (e.g., the U.S. or Europe) to highlight regional differences or similarities.

Response 10: We thank the reviewer for this comment. We have added a paragraph in the Discussion section addressing this point.

Comments 11: Discuss how the identified risk factors align or contrast with findings from other studies (e.g., references [6], [7], [20]).

Response 11: We thank the reviewer for this comment. A paragraph has been added to the Discussion describing practices reported in other studies that have been effective in reducing the risk of PRRSV dissemination.

4. Response to Comments on the Quality of English Language

Point 1: Needs english language editing, grammer and typos. Making the language run flow and easy.

Response 1: The English language throughout the manuscript has been improved by MDPI English editing service (Author Services ID: english-99589).

Reviewer 2 Report

Comments and Suggestions for Authors

The authors have investigated associations between various risk factors for pigs testing positive to porcine reproductive and respiratory syndrome virus by RT-qPCR.

The introduction provides the information need to understand the study and is supported by relevant references. The study aims are clearly stated.

The materials and methods are described in sufficient detail to enable replication of the study. The samples analysed in the study were collected and have been reported in a previous publication.

The results are clearly presented, and the main text reflects what is shown in the tables.

The discussion is mostly relevant and places the results of the study in the context of the relevant research. However, there are some results that warrant further discussion. In Table 2 there are some odds ratios where the 95% confidence intervals are extremely wide, for example “access”, “circo”, “ery” and “wild”, have the authors considered what confounders or uncontrolled factors might underpin these imprecise values?

The conclusions are consistent with the stated aims of the study.

Comments and suggestions

Line 41 suggest revision “that affects most pig populations worldwide”

Line 134 suggest revision “Reverse transcriptase quantitative real time PCR (RT-qPCR) analysis was used”

The use this abbreviation through the manuscript.

I think this is the most accepted and accurate term for this assay.

Line 188 suggest revision by deleting “from the highest to lowest frequency”, it is redundant.

Line 207 I would suggest the authors define what type of vaccines that may have been used or known to be available in the materials and methods. It is not until later in the manuscript that the use of MLVs is mentioned.

Lines 339-342 How do the authors interpret this alternative analysis? Do they think that it could be a consequence of them detecting the vaccine strains? Do the authors know the interval between vaccination and the time of sampling? Why are these results not shown in Table 2?

Author Response

Thank you very much for taking the time to review this manuscript. Please find detailed responses below and the corresponding revisions/corrections highlighted/in track changes in the re-submitted files.

Comments 1: In Table 2 there are some odds ratios where the 95% confidence intervals are extremely wide, for example “access”, “circo”, “ery” and “wild”, have the authors considered what confounders or uncontrolled factors might underpin these imprecise values?

Response 1: We thank the reviewer for this insightful comment. The wide confidence intervals (CIs) observed for some odds ratios (ORs) may result from the distribution of observations across strata and the outcomes of the diagnostic tests. A wide CI indicates that the true OR is difficult to estimate and could lie anywhere within this range. This is expected in cross-sectional field studies, such as ours, where the distribution of observational units is not controlled and reflects the actual dynamics of production and infection at a given point in time.

Although the precision of these ORs is limited, the results offer a valuable overview of the epidemiological situation under real-world conditions. Future longitudinal studies, including case-control designs, would allow for more precise and robust estimates of the effect of risk factors on PRRSV viremia.

Comments 2: Line 41 suggest revision “that affects most pig populations worldwide”

Response 2: We appreciate the reviewer’s suggestion. To improve accuracy, we have revised the sentence to better reflect the global distribution of PRRS while acknowledging countries where the virus is not present.

Comments 3: Line 134 suggest revision “Reverse transcriptase quantitative real time PCR (RT-qPCR) analysis was used” The use this abbreviation through the manuscript. I think this is the most accepted and accurate term for this assay.

Response 3: We thank the reviewer for this valuable suggestion. We have standardized the term Reverse transcriptase quantitative real-time PCR (RT-qPCR) throughout the manuscript to ensure consistency and accuracy.

Comments 4: Line 188 suggest revision by deleting “from the highest to lowest frequency”, it is redundant.

Response 4: We appreciate the reviewer’s observation. To improve clarity and conciseness, we have removed the phrase “from the highest to lowest frequency” from the manuscript and slightly revised the sentence to enhance readability.

Comments 5: Line 207 I would suggest the authors define what type of vaccines that may have been used or known to be available in the materials and methods. It is not until later in the manuscript that the use of MLVs is mentioned.

Response 5: We thank the reviewer for this suggestion. The information regarding the type of vaccine (MLV) and the vaccination protocol used on the farms has now been added to the Materials and Methods, “Samples” section. This ensures that readers are informed about vaccination practices prior to the results section.

Comments 6: Lines 339-342 How do the authors interpret this alternative analysis? Do they think that it could be a consequence of them detecting the vaccine strains? Do the authors know the interval between vaccination and the time of sampling? Why are these results not shown in Table 2?

Response 6: We thank the reviewer for this valuable comment. The paragraph describing the association between vaccination and viremia has been thoroughly rewritten to improve clarity and flow. Additionally, the relevant data have been incorporated into Table 2 of the Results section to provide a clear overview.

Although the exact interval between vaccination and sample collection is unknown, pig flow and vaccination management on the farms are recurrent and standardized. Therefore, the observed viremia in vaccinated pigs is consistent with vaccine-induced viral circulation. The reciprocal values of the odds ratios (ORs) were calculated to allow an alternative interpretation of the effect of vaccination on the probability of viremia, providing technical insight into this phenomenon.
